# Detecting sentiment dynamics and clusters of Twitter users for trending topics in COVID-19 pandemic

**Md Shoaib Ahmed◉ⓢ\*, Tanjim Taharat Aurpaⓢ, Md Musfique Anwarⓢ**

Department of Computer Science and Engineering, Jahangirnagar University, Savar, Dhaka, Bangladesh

ⓢ These authors contributed equally to this work.
\* shoaibmehrab011@gmail.com

**Data Availability Statement:** The sample data used in this study can be found on figshare at https://figshare.com/articles/dataset/Covid19RelatedTweets/14735972.

## Abstract

COVID-19 caused a significant public health crisis worldwide and triggered some other issues such as economic crisis, job cuts, mental anxiety, etc. This pandemic plies across the world and involves many people not only through the infection but also agitation, stress, fret, fear, repugnance, and poignancy. During this time, social media involvement and inter-action increase dynamically and share one's viewpoint and aspects under those mentioned health crises. From user-generated content on social media, we can analyze the public's thoughts and sentiments on health status, concerns, panic, and awareness related to COVID-19, which can ultimately assist in developing health intervention strategies and design effective campaigns based on public perceptions. In this work, we scrutinize the users' sentiment in different time intervals to assist in trending topics in Twitter on the COVID-19 tweets dataset. We also find out the sentimental clusters from the sentiment categories. With the help of comprehensive sentiment dynamics, we investigate different experimental results that exhibit different multifariousness in social media engagement and communication in the pandemic period.

## 1 Introduction

People's involvement in the online social network (OSN) has increased during the COVID-19 pandemic, as regular activities move online. Numerous uses of OSN (e.g., people use OSN for expressing their opinion, communicating with family members, online meetings, etc.) are showed up at this time. Like other OSN, the use of popular microblogging service Twitter has also been impacted. It becomes a popular media for the leaders to communicate with general people and make them aware of public health during this health crisis [1]. So, people usually spend more time on Twitter, and users are more active than at any other time. Their involvements increase during the lockdown period to get the latest news on COVID-19. At the same time, they share their opinions and feelings with their friends through it. As a result, analysis of Twitter data draws vast attention from researchers in this pandemic.

**Funding:** The authors received no specific funding for this work.

**Competing interests:** No authors have competing interests.

Sentiment analysis is a technical study about people's emotions, opinions, and attitudes [2]. It is an effective way to measure people's thoughts on particular topics. Moreover, sentiment analysis can convey various impacts on society in several ways. Different types of mental anxieties arise in this pandemic situation, and all those mental conditions can be summarized through sentiment analysis. We can quickly determine the extensive state of depression and panic disorder of persons in a society or community from the sentiment analysis result. We need to apply different virtual depression optimizers in those depressed persons to bring some positive ramifications to society. Again, the success of many applications like recommendation systems depends on the sentiments of social users. Sentiment analysis for active users is a more efficient way to track public opinion. In the coronavirus pandemic, these types of research have significant contributions to help government and policymakers. Authors in [3] analyze Indian people's sentiment during corona lockdown. They used some popular hashtags for measuring positivity and negativity in people.

People concentrate on many different topics during this whole pandemic period. Some people posted tweets about the COVID-19 tests and deaths. Again, some people focused on job cuts, online education, or politics. Besides the new topics arrival among people, many different thoughts regarding those topics are shown in this pandemic situation. In [4], authors determine top trending topics using hashtags for detecting COVID-19 conspiracy theories. Another work [5] detected trending topics and clustered them using the $k$-mean clustering algorithm. So, the determination of trending sub-topics at different time windows is essential to understand the public's changing interests properly.

Our work includes the concept of analyzing active users' different sentiments, such as positive, negative, and neutral sentiments at a particular time interval for trending topics related to COVID-19. This work concentrates on people's positive, negative and neutral sentiments on top-$k$ trending sub-topics in Twitter related to COVID-19. We also track the changes that occurred in top trending topics in Twitter and user's sentiment. The main contributions of our research are summarised below:

- Propose a model that lists top-$k$ trending topics in Twitter due to COVID-19 pandemic at a different time interval.

- We are modeling and evaluating users' sentiments towards different topics of a given query.

- Modeling the sentiment dynamics of different topics.

- Detection of sentiment clusters and tracking their changes for top-$k$ trending topics over time.

We have accomplished this work as an extensive version of our extended abstract that appeared at [6]. The significant key points of our additional contributions in this journal version are listed below:

I.  We cluster the Twitter users based on their sentiments on different topics related to COVID-19.

II.  We model the degree of topical activeness of the users according to the rank of the topics of a given query.

III.  We revise the existing algorithm to list top-$r$ users according to their overall activities related to top-$k$ trending topics.

IV.  We conduct our experiment on a new dataset that contains COVID-19 related tweets. We collect those tweets with real-time Twitter lookup API and prepare them according to our requirements.

V. The COVID-19 outbreak results in an overwhelming amount of information on different topics, and also users' sentiments vary quickly. As a result, we consider a non-overlapping time window with shorter time intervals to monitor social users' sentiments.

VI. In most cases, tweets are very informal, extremely noisy, and also contain grammatically incorrect phrases. To improve the quality of data, we apply a set of pre-processing steps such as Tokenization, Lemmatization, Stemming, Sentence Segmentation, etc. for performance enhancement.

VII. We consider a self-regulating topic modeling approach known as T-LDA (Twitter-Latent Dirichlet Allocation) [7] to detect the topic from a tweet.

## 2 Related work

Rajesh et al. [8] scrutinized Tweets related to the coronavirus to get out the appropriate and most accurate with minor misinformation spread. Here, applied only the LDA (Latent Dirichlet Allocation) analysis to find out the negative sentiments dominated the tweet as expected as the virus highly contagious that was clear from the sentiment analysis significantly depends on some words. This work only shows the negative sentiment just from some particular topics without analyzing any model in time intervals and devoid of any sentiment model and analysis. Jim Samuel et al. [9] presented an issue surrounding public sentiment leading to the testimony of growth in fear sentiment and negative sentiment. This approach does not examine the change of sentiment aloft time. An evolving method [10] illustrates the sentiment analysis country-wise related to COVID-19. The author evokes sentiments from tweets only with the judgment of some growing keyword about coronavirus of examining the top trending topics over time. They also discuss just positive and negative sentiments. This approach does not consider any extensive topical model (ex. T-LDA) and neutral sentiment. Yin et al. [11] introduced a structure to study the topic and sentiment dynamics due to COVID-19 from extensive Twitter posts. A recent proposal [12] to analyze social media (micro-blogging like as Twitter called Weibo) data in the early stage of COVID-19 in China and proposed a topic extraction and classification model. The opinion's appearance showed that the topic's approach is stable and viable for understanding public opinions. Moreover, they showed the statistical results of the percentage of first-level topics of COVID-19. A machine learning-based sentiment analysis [13] introduced a hybrid approached to find out the sentiments on regular tweets with polarity calculations. The polarity score is measured from a score range of -1 to 1 based on words used and then used three sentiment analyzer W-WSD, TextBlob, and SentiWordNet. Those analyzers are then validated with the Waikato Environment for Knowledge Analysis (Weka) to measure the best result. Pandey et al. [14] proposed a metaheuristic method depend on K-means and cuckoo search. This method is applied to the different tweeter datasets to determine the optimum cluster-heads in terms of sentiment. It is also compared with differential evolution, particle swarm optimization, cuckoo search, improved cuckoo search, two n-grams, and gauss-based cuckoo search.

A clustering-based approach on sentiment analysis is proposed by Gang [15] where they accosted a weighting method called term frequency-inverse document frequency (TF-IDF) on document-based content. Over the two existing forms of propositions, they listed a competitive advantage, one is allegorical methods, and another is supervised learning methods. They used the simple k-mean clustering algorithm to find the positive and negative categories of clusters. An SVM classifier combined with a cluster organization provided better classification accuracies than a stand-alone SVM to control the impressions, feelings, and biases presented in the source material to assess tweet sentiment analysis [16]. They used an algorithm called

C3E-SL in their analysis, capable of combining classifier and cluster assemblies. This algorithm will improve tweet classifications from clusters' additional details, assuming the same class-mark is more likely to be shared by similar instances from the same clusters. Shreya et al. [17] suggested a study that came from various clusters that belong to polarity wise and subjective wise with sentiment ratings. The sentiment scores are assessed here using Afinn and TextBlob. Therefore, they used extensive data, calculating the Euclidean distance in less time and using the K-means clustering algorithm technique. An extensive approach [18] to find out the appearance of clustering techniques on document sentiment analysis. In their first approach, they showed two types of notices. The first one is a good performance, and the second one is the poor performance when applying the K-means-type clustering algorithm on balanced and unbalanced datasets, respectively. To avoid this problem, they designed a weighting model that worked well on both unbalanced and balanced datasets that were better than the conventional weighting model. Feng et al. [19] researched clustering methods on standard blog posts and got natural emotions from web blogs by topics or keywords, which is a typical approach. A novel approach based on Probabilistic Latent Semantic Analysis (PLSA) is performed. An emotion-oriented clustering technique is proposed to find common emotions affirming the connection of fine-grained sentiment between blogs and blog posts. Farhadloo et al. [20] proposed a score representation with aspect level sentiment identification. This identification is based on positiveness, neutralness, and negativeness. This process is designed with a 3-class SVM classifier to determine feature sets according to a 3-dimensional representation (positive, negative, and neutral). To improve clustering results, authors utilized a bag of nouns (BON) rather than a bag of words (BOW).

## 3 Preliminary and proposed framework

We introduce some relevant concepts before defining the problem statement. Then we give an overview of our proposed framework.

**Social Graph**: We model the Twitter network as a social graph $G = (U, E, \mathcal{T})$, where $U$ is the set of nodes (users), $E$ is the set of connections or virtual social relationships among the Twitter users (such as the *following* relationships in Twitter), and $\mathcal{T} = \{T_1, T_2 ..., T_m\}$ is the set of topics discussed by the social users $U$ [21].

**Topic**: A topic is a collection of the most representative words for that topic. For example, *politics* topic has words like election, government, democratic, parliament, etc. about politics [22, 23].

**Social Stream**: A social stream $S$ is a continuous and temporal sequence of the tweets posted by the social users $U$.

**Query**: An input query $Q = \{\mathcal{T}_q\}$ consisting top-$k$ trending Topics $\mathcal{T}_q = \{T_i, T_{i+1} ..., T_k\}$ at a particular time interval.

**Overlapping Time Window**: A window of a predefined length *len* is moved over the social stream $S$ and specifies the intervals to analyze. Let $\Gamma = <t_1, t_2, \ldots, t_n>$ be a sequence of points in time, $I_m$ an interval $[t_{i-len}, t_i]$ of *len*, where $0 < len \leq i$. We partition $\Gamma$ into set of equal-length intervals denoted as $\mathcal{I} = \{I_1, ..., I_m\}$. We consider an *overlapping window* partially overlaps with the prior window. The degree of overlap is controlled by the parameter $\Delta t$ [24].

**Topical Involvement Score**: For each user $u_i \in U$, we compute her involvement score towards the query $Q$ in a time interval $I_m$ using Eqs 1 and 2 which measures $u_i$'s relative participation compared with the most active users at that time interval $I_m$.

$$\sigma_{(u_i, Q, I_m)} = \frac{\psi_{(u_i, Q, I_m)}}{max_{u_z \in U_{(Q, I_m)}} \{\psi_{(u_z, Q, I_m)}\}} \tag{1}$$

$$\psi_{(u_z,Q,I_m)} = \sum_{l=1}^{k} (k+1-l) \cdot \kappa_{(u_z,Q,I_m)} \qquad (2)$$

where $\kappa_{(ui, Q, I_m)}$ indicates the total number of tweets posted by $u_i$ related to $l$th number topic in $Q$ at $I_m$.

Our proposed approach has three stages as presented in Fig 1. Firstly, the pre-processing is performed to remove irrelevant data from the social stream $S$. Secondly, we apply the topic modeling method on the cleaned data to infer the latent topics and then select top-$k$ trending topics. Then we apply our proposed algorithm to the processed social streams to find top-$k$ trending topics and users' involvement scores. Finally, we detect involved uses' sentiment dynamics and clusters of top-involved users at different time intervals.

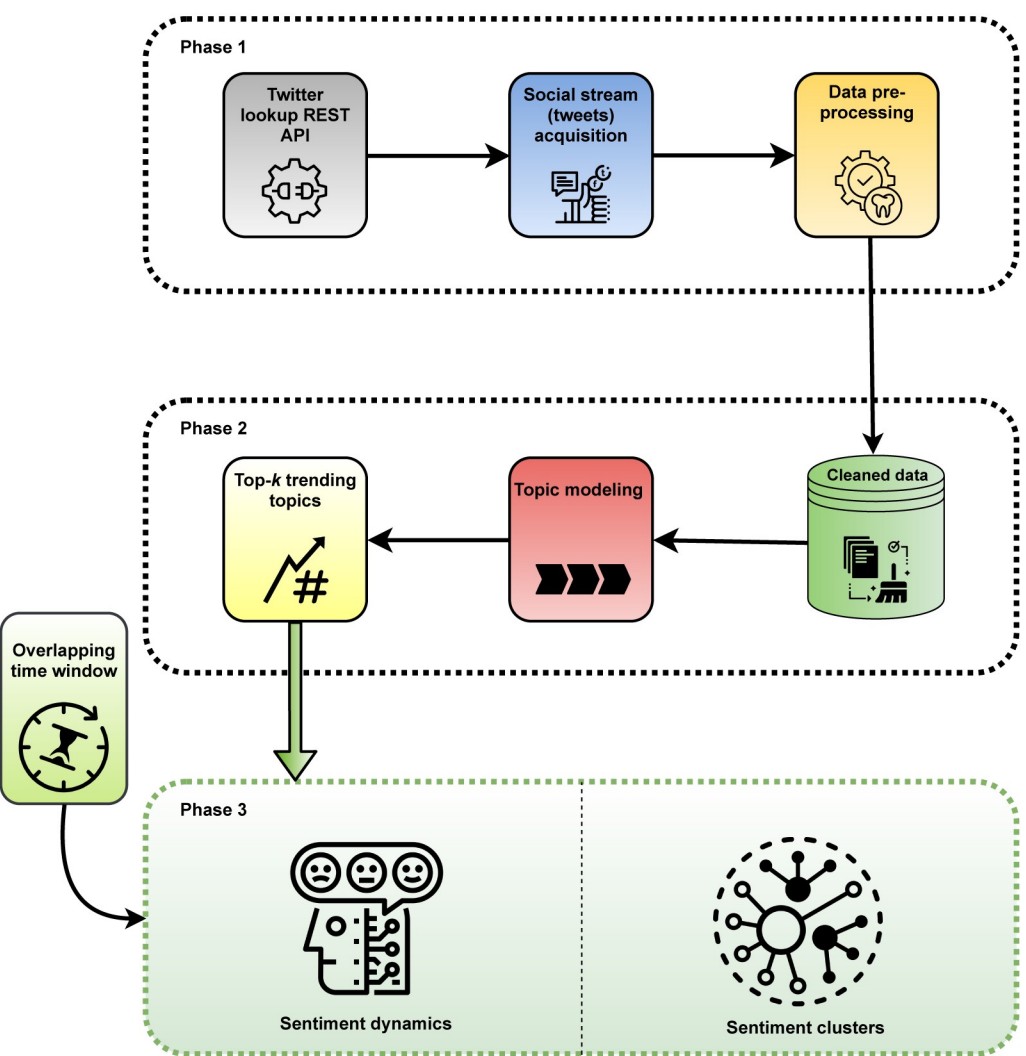

**Fig 1. The workflow of the proposed framework (a methodical diagram representing the entire process from data collection to topic modeling and find out the sentiment dynamics and clusters).**

### 3.1 Data pre-processing for topic detection

In general, tweets are informally written and often contain grammatically wrong sentence formations with misspellings and non-standard words. Tweets also contain numerous non-standard forms (e.g., comeee for Come, goooood for good), informal abbreviations (e.g., tmrw for tomorrow, lemme for let me, wknd for weekend), phonetic substitutions (e.g., gdn8 for good night, 4eva for forever, 2day for today), etc. For removing those, we follow some steps to lead the standardization for our next stages. In the first stage, we remove the noise entities such as HTML tag, Stop words, Punctuations, White Spaces, URLs, etc. The next stage is text normalization like as Tokenization, Lemmatization, Stemming, Sentence Segmentation, etc. Finally, word standardization gives us cleaned texts. To improve the quality of our tweet corpus and the fulfillment of the consequent steps, mentioned normalization of the tweets through linear substitution of lexical variants with their conventional forms proposed by Han et al. [25]

### 3.2 Topic detection from social stream

The use of hashtags (for example #coronavirus, #StayHome) to point out a tweet's topic is common on Twitter. However, neither every tweet contains hashtags, nor hashtags have been written by following any rule. Thus, tracking hashtags rarely leads to the exact topic. Another topic modeling approach T-LDA (Twitter-Latent Dirichlet Allocation) [7], is a popular way of inferring topics on Twitter. It is a textual analysis tool that deals with microblogs like tweets. Tweets are limited to 140 characters, and within this limitation, a single tweet can refer to a single topic. This restricted characteristic of tweets intercepts traditional text mining tools in their successive execution. T-LDA resolves this issue and potentially works with tweets.

Twitter LDA has been implemented based on the following assumptions.

- Assuming there are T topics on Twitter and each topic $t$ is generated from background word distribution $\theta_B$ and topic word distribution $\theta_t$. Latent value $y$ dominated by Bernoulli distribution $\pi$. identifies a word w to be a background word ($y = 0$) or a topic word ($y = 1$).

- $\Phi_u$ represents a user $u$'s topic of interest. It also determines the assignment of topic $t$ for each word in tweets posted by $u$.

- $\alpha_d, \beta_d, \gamma_d$, and $\lambda_d$ are the parameters of the Dirichlet prior on $\Phi_u, \theta_t, \pi$ and $\theta_B$ respectively.

- z is the determined topic for a tweet [26].

Fig 2 shows the graphical representation of T-LDA. Table 1 shows the word distribution for top-$k$ topics (k = 3) in different time intervals from 23$^{rd}$ March, 2020 to 31$^{st}$ March, 2020.

### 3.3 Top-$k$ trending topics from social stream

In our proposed model, we set the value of the query $Q$ at each time interval $I_m$ as the top-$k$ trending topics related to COVID-19 at that $I_m$. We define trending score ($\Lambda_{(Ti, I_m)}$) for each topic $T_i$ according to Eq 3:

$$\Lambda_{(T_i, I_m)} = \alpha \times N_{T_i} + (1 - \alpha) \times U_{T_i, I_m} \tag{3}$$

Where $N_{T_i}$ indicates the total number of tweets related to topic $T_i$ and $U_{T_i, I_m}$ represents the number of unique Twitter users who posted tweets on $T_i$ at time interval $I_m$. The parameter $\alpha \in [0, 1]$ balances the above two factors.

Table 2 shows how the changing value of alpha can effect the top trending topics at a particular time window. In this table top-$k$ topics for three different values of $\alpha$ at two different time

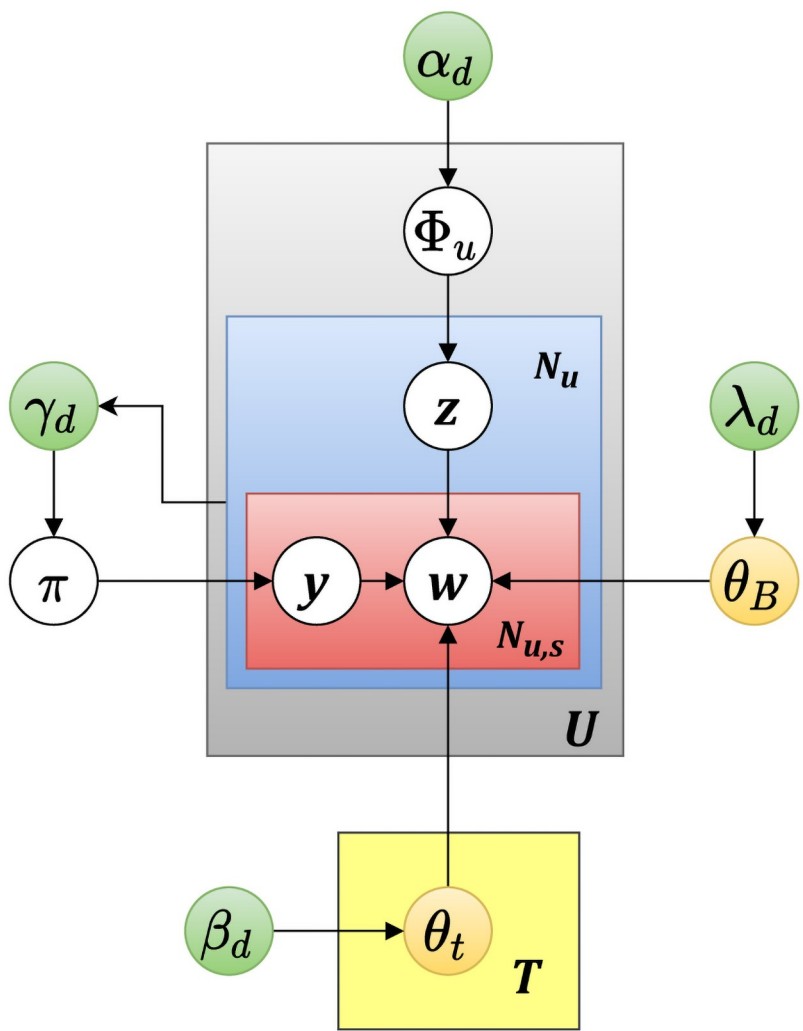

**Fig 2. Graphical representation of Twitter-LDA model.**

**Table 1. Topic word distribution for top-*k* trending topics in Twitter.**

| News | Health | COVID-19 Test | Lockdown |
|---|---|---|---|
| Press | Hospital | Tested | Home |
| Live | Died | Positive | Locked |
| Conference | Ventilator | Confirmed | Safe |
| CNN | Drug | Report | Distancing |
| Fox News | Patient | Negative | Staying |
| Briefing | Medical | Update | Friend |
| World | Nurse | Total | Hope |
| Staff | Equipment | Number | Save |
| Time | Supply | Covid | Family |
| Medium | Doctors | Rate | Work |

**Table 2. Top-*k* trending topics for different values of *α*.**

| *α*↓ | *I_a*(25/03-28/03) | | | *I_b* = (26/03-29/03) | | |
|---|---|---|---|---|---|---|
| 0.3 | News | Health | Economy | News | Health | Politics |
| 0.4 | News | Health | Lockdown | Health | News | COVID-19 Test |
| 0.5 | News | Health | Lockdown | News | Health | COVID-19 Test |

**Table 3. Top-*k* = 3 trending topics in different *I_m* (we consider seven time intervals where *len* = 3 and Δ*t* = 1).**

| *I_1* (23/3–25/3) | *I_2* (24/3–26/3) | *I_3* (25/3–27/3) | *I_4* (26/3–28/3) | *I_5* (27/3–29/3) | *I_6* (28/3–30/3) | *I_7* (29/3–31/3) |
|---|---|---|---|---|---|---|
| News (19%) | News (20%) | News (20%) | News (20%) | News (21%) | Health (22%) | Health (23%) |
| Health (16%) | Health (18%) | Health (20%) | Health (18%) | Health (21%) | News (21%) | News (22%) |
| Lock down (16%) | Covid Test (17%) | Covid Test (19%) | Covid Test (17%) | Covid Test (20%) | Politics (17%) | Politics (16%) |

intervals ($I_m$) $I_a$ = 25/03 − 28/03 and $I_b$ = 26/03 − 29/03. We consider the length of each $I_m$ for four days and shift this $I_m$s for Δ*t* = 1 day.

We use Eq 3 ($α$ = 0.5) on different time intervals and detect top-*k* (*k* = 3) trending topics on Twitter. We consider seven-time intervals starting from 23rd March 2020 to 31st March 2020. Each of these time interval's length is 3 days and we shift them by Δ*t* = 1 days. We also measure how much of these trending topics are discussed by users at those time intervals. Table 3 shows the percentages of top-*k* = 3 trending topics that indicate it's popularity among users.

Fig 3 represents the percentage of topics discussed by users at different time windows. It is a heatmap where different blue shades are used to indicate the trendiness of topics at various $I_m$. We consider $α$ = 0.5 to determine the trendiness of topics. We depict this heatmap using seven

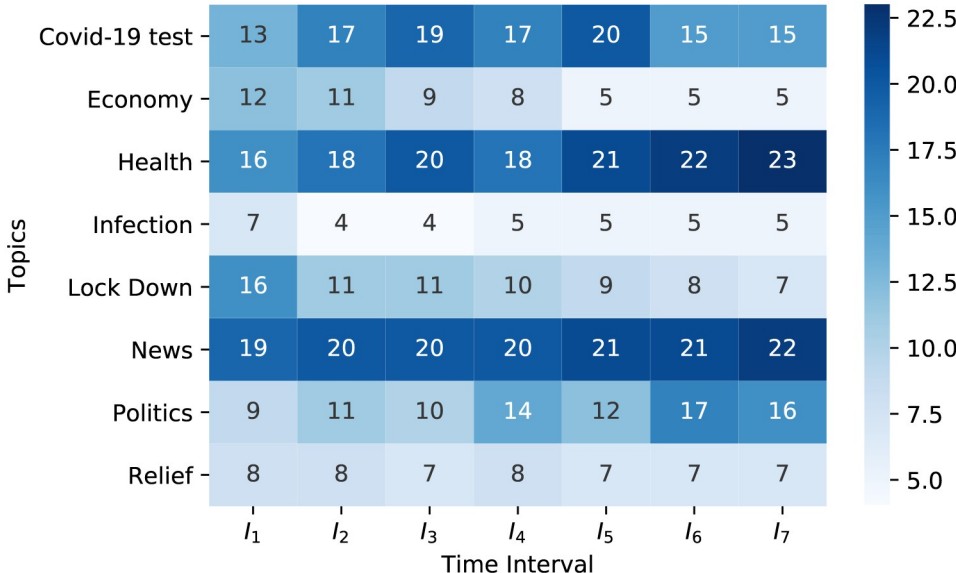

**Fig 3. Heatmap representing the percentage of each topic discussed among users at various time interval $I_m$** (displays the top eight discussed topics at seven different time-intervals).

different time windows from 23/03/2020 to 31/03/2020. Each of them has a length of 3 days and is shifted by one day. We pick the top eight trending topics *(Health; News; COVID-19 test; Lockdown; Economy, Politics, Relief, and Infection)* on Twitter to visualize the distinction of the percentage of those topics discussion rate among users.

### 3.4 Users' involvement Detection Algorithm

We develop an algorithm that can detect top-$k$ trending topics before determining the top involved users at a particular time.

**Algorithm overview**. The algorithm, called `Query Algorithm`, identifies top-$k$ topics from social stream $S$ at each time interval $I_m$ through procedure TOP_K_TOPICS (line 9-17) at first. It enumerates the trending score $\eta_{(T_j, I_m)}$ for each topic $T_j$ and adds that score to a priority queue of size $k$ (line 11-16). Then it returns the top-$k$ topics based on their trending scores. Next, the algorithm finds the set of users $U_{I_m}^Q$ from $U$ for a given $Q$ at each time interval $I_m$ and then computes users' involvement score $\sigma_{(ui, Q, I_m)}$ (line 3-6). Finally, the users are sorted by their involvement scores, and the proposed algorithm returns the top 20 users as output (line 7-8).

**Algorithm 1** `Query Algorithm`

**Require:** $G = (U, E, \mathcal{T}), \mathcal{I}, Q, S, k, \alpha$
**Ensure:** top-$r$ active users $U_{I_m}^r$

```
1: for each I_m ∈ I do
2:    Q ← TOP_K_TOPICS(S, I_m, α)
3:    select U_{I_m}^Q from U ▷ each u_i ∈ U has to post certain number of
tweets related to Q
4:    for each u_i ∈ U_{I_m}^Q do
5:       compute σ_(ui, Q, I_m)
6:    end for
7:    Sort the list U_{I_m}^Q according to σ_(ui, Q, I_m)
8:    top-r active users U_{I_m}^r at each time interval I_m
9:    Procedure TOP_K_TOPICS(S, I_m, α)
10:    P ← PriorityQueue(k)
11:    for each T_j ∈ T do
12:       compute the total number of tweets |ψ_(ui, Q, I_m)|
13:       generate user frequency matrix U_{T_j, I_m}
14:       compute η_(Tj, I_m)
15:       P.add(η_(Tj, I_m))
16:    end for
17:    return top-k results from P
18: end for
```

### 3.5 Sentiment identification from social stream

For sentiment identification, we use VADER (Valence Aware Dictionary and Sentiment Reasoner) [27] is a lexicon and rule-based sentiment analysis appliance that precisely harmonizes to sentiments expressed in social media. VADER is open-source and licensed under the MIT available in GitHub. It is the rule-based sentiment analysis engine that carries out the grammatical and syntactical rules. In addition, it recognizes the intensity of sentiment in sentence-level text.

Our processed social streams pass through this engine for the analysis of sentiments and give a score. The scoring formulation is given below:

- The *compound* score ($\varrho$) is calculated by summing the valence scores of each word in the lexicon, adjusted according to the rules, and then normalized to be between $\Upsilon_{max}$ and $\Upsilon_{min}$. It is

suitable for a single uni-dimensional measure of sentiment for a given sentence.

Where, $\Upsilon_{min} = -1$ = most extreme negative and $\Upsilon_{max} = 1$ = most extreme positive. Here we take the graded thresholds for classifying sentences as either positive, neutral, or negative. Typical threshold values are:

- **positive sentiment**: $\varrho \geq 0.05$

- **neutral sentiment**: $\varrho > $ -0.05 and $\varrho < 0.05$

- **negative sentiment**: $\varrho \leq $ -0.05

- The *pos, neg*, and *neu* scores are the proportion of each category and the multidimensional measures of sentiment for a given sentence.

From Table 4, there we look at three columns. The first column is the social streams (tweets), the second column is polarity, where we observe the value of different sentiments between $\Upsilon_{max}$ and $\Upsilon_{min}$ after applying VADER. Then eventually, we classify the social streams as either positive, negative, or neutral.

## 4 Experimental evaluation

In this section, we estimate the performance of our algorithm on a real Twitter dataset. We perform all experiments on an AMD Ryzen 7 3700U with Radeon Vega 10 Gfx (8 CPUs), 2.3 GHz Windows 10 PC with 32 GB RAM and 512GB NVME M.2 SSD.

### 4.1 Data set

We collect COVID-19 related tweets through Twitter lookup API's endpoint that contains 100 million tweets with 10,000 users from 23 March 2020 to 31 March 2020.

### 4.2 Performance evaluation measure

We consider two performance evaluation measures, one is entropy, and another one is semantic cohesion.

Entropy measures with the Equation of 4 and 5 that betokens the randomness of topics discussed in clusters.

$$entropy(\{\mathcal{C}j\}_{j=1}^{r}) = \sum_{j}^{r} \frac{|U(\mathcal{C}_j)|}{|U|} entropy(\mathcal{C}_j) \tag{4}$$

**Table 4. Sample tweets with sentiment polarity by VADER.**

| Social stream | Polarity | Result |
|---|---|---|
| Emerging markets have limited power to tackle recession #economy #businessnews #coronavirus #emerging #emergingmarkets #healthcareindustry | Neg: 0.299, Neu: 0.701, Pos: 0.0, Com.: -0.5719 | Negative |
| My Mom's a nurse and just tested positive for COVID-19. The caregiver is now the patient. Stay home for all the brave | Neg: 0.0, Neu: 0.562, Pos: 0.438, Com.: 0.7906 | Positive |
| Under Armour Manufacturing Face Masks For Hospital Workers Amid Coronavirus Pandemic #economy #armour #coronavir. . . https://t.co/J7Rkz8WMjw | Neg: 0.367, Neu: 0.412, Pos: 0.221, Com.: -0.2019 | Neutral |

$$entropy(\mathcal{C}_j) = -\sum_{i=1}^{n} p_{ij} log_2 p_{ij} \tag{5}$$

Here, $\dfrac{|U(\mathcal{C}_j)|}{|U|}$ is the weighted probability of a user in cluster $\mathcal{C}_j$ for discussing a trending topic. $p_{ij}$ is the percentage of active users for that topic in the cluster. $entropy(\{\mathcal{C}_j\}_{j=1}^{r})$ measures the weighted entropy considering all topics over all the ($r$) clusters. Usually, a good topical cluster should have a low entropy value.

Semantic cohesion is measured with the following Equation from 6 to 8. For this purpose, we find out the main topic of activity of each user $u_i$ according to Eq 6.

$$\lambda_{(u_i, I_m)} = freqmax_Q ACTS(u_i, \psi_{u_i}) \tag{6}$$

Then, the most recurrent topic in a cluster $\mathcal{C}_j$ at time interval $I_m$ defines with the Eq 7.

$$\lambda_{(\mathcal{C}_j, I_m)} = freqmax_Q \lambda_{(u_i, I_m)} \tag{7}$$

Finally, we find the semantic cohesion (expertness of cluster) denoted as $\rho_{(\mathcal{C}_j, I_m)}$ for a particular topic $T_j$ at time interval $I_m$ with Eq 8.

$$\rho_{(\mathcal{C}_j, I_m)} = \frac{\#\{u_i \in \mathcal{C}_j, \ \lambda_{(u_i, I_m)} = \lambda_{(\mathcal{C}_j, I_m)}\}}{|\mathcal{C}_j|} \tag{8}$$

## 4.3 Experimental results

In this section, we have mentioned the findings of our experiments. Firstly we detect top-$k$ = 3 trending topics and identify the involved users for those topics using our query algorithm. Then we determine their sentiments. Based on these experiments, we make different types of observations regarding users' sentiments.

We consider our Table 3's topics set in each time interval for further experiment. Our query is a set of topics and we fixed $k$ = 3 and $\alpha$ = 0.5 for determining these topic sets. We consider all negative, positive, and neutral tweets for different time windows. We demonstrate the result not only for our query $Q$ = {*set of top-k = 3 trending topics*} but also for individual topic in the query set. We sketch bar diagrams in Figs 4–6 to represent percentages of positive, negative and neutral tweets posted by our users on a particular time interval for News, Health and COVID-19 test. These three topics appear again and again as top trending topics. *Sea green, Coral red* and *Royal blue* bars are indicating the positive, negative, and neutral tweets, respectively.

After that, we concentrate on the most involved users' sentiments towards COVID-19 related subtopics. We determine users' involvement scores for our query topics (top trending topics from Table 3) at different time intervals. We identify the top 20 involved users at each time interval.

Our research has been accomplished with recent year's tweets related to COVID-19. To preserve the privacy of users, we replace some alphabets with '*' in usernames. Table 5 shows the sentiment dynamics for the top 20 involved users at each $I_m$. To measure overall sentiment dynamics, we sum up users' sentiment scores towards the query topics. In that table, we can see that the top 20 involved users' list is changing over time.

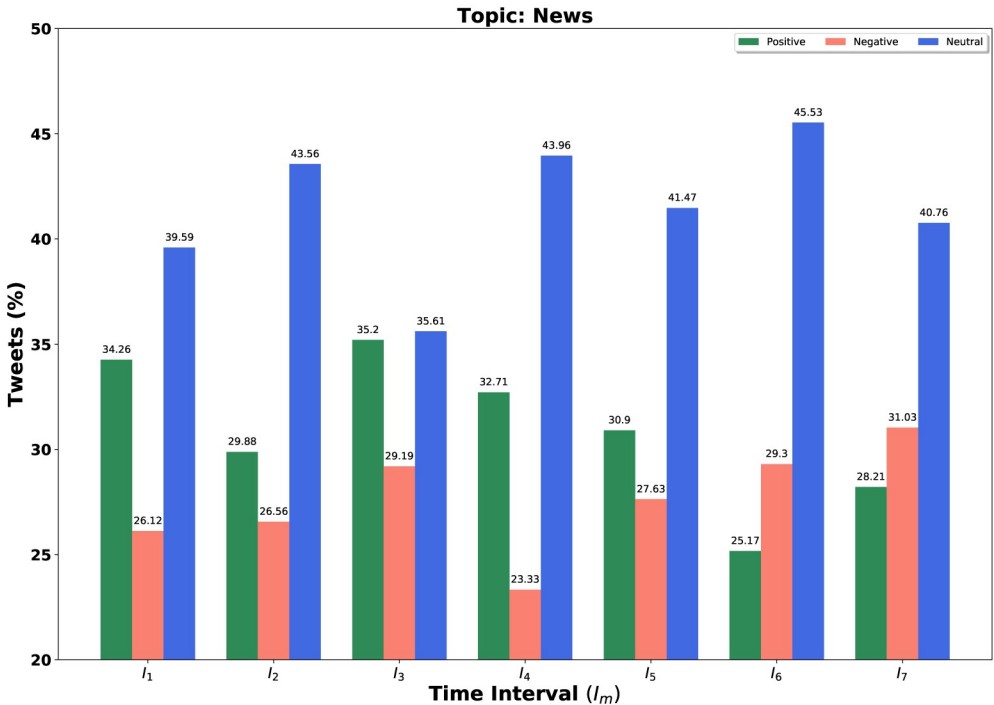

**Fig 4. Overall sentiment dynamics on Twitter at different time intervals $I_m$ for topic news.**

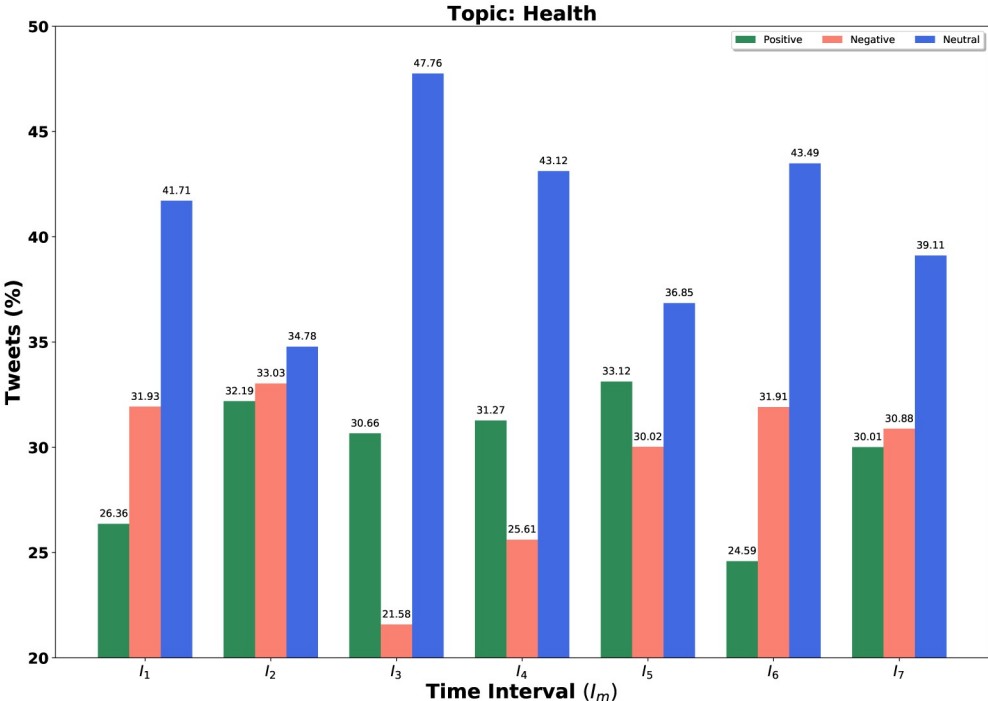

**Fig 5. Overall sentiment dynamics on Twitter at different time intervals $I_m$ for topic health.**

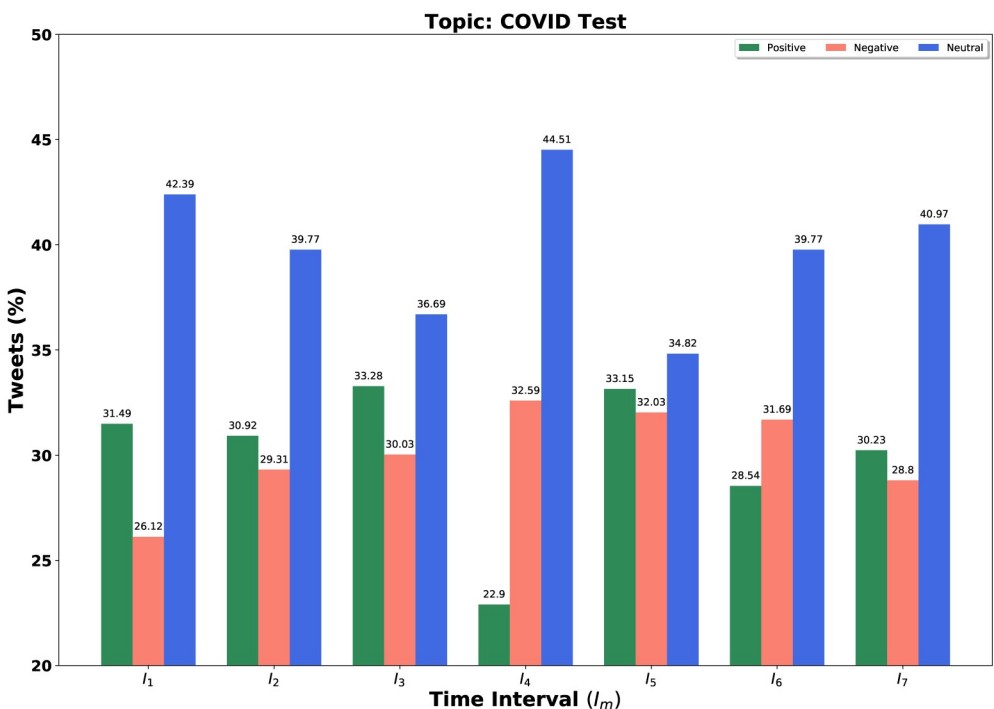

**Fig 6. Overall sentiment dynamics on Twitter at different time intervals $I_m$ for topic COVID-19 test.**

The reason behind this is that user's interests and their involvements in trending topics vary over time. Another remarkable fact in this table is the change of users' sentiments over time. Users who remain in the top 20 on the next $I_m$ have different sentiment scores.

Here we analyze some users' sentiment dynamics with their involvement below:

- PK**17 is highly involved in each $I_m$. He has positive sentiments in $I_1, I_2$ and $I_3$ and diverts to negative sentiments from $I_4$ to $I_7$. More highly involved users like PK**17 have different sentiment dynamics at each $I_m$.

- Other types of users like ma**te remain in the top 20 at some time windows. But also drop from the involvement list at the next or previous time windows. ma**te is a top involved user in $I_1$, $I_2$, $I_3$, but vanishes from the list after $I_4$. These users have various sentiment dynamics at a particular $I_m$.

- Some users suddenly appear in the top involved list, who have no existence in the list previously (e.g., jg**00). User jg**00 is not one of the top involved users at $I_1$ and $I_2$, while he is scoring top at the next three $I_m$. This user has non-identical sentiment scores over time.

We analyze the top 20 users and bring out ten users whose average involvement in seven $I_m$ is greater than other users. We track the changes in their sentiment dynamics. In Fig 7 we sketch these 10 users' sentiments. Here we can observe that users have different types of sentiment scores at different time windows. Even for some users, their sentiment dynamics changes from positive to negative or neutral. After another shift, it is changing into positive again. This heatmap provides clear visualization of sentiments' change over time for a particular user.

We also determine sentimental clusters based on the users' sentiment scores. We find cluster $C_{Pos}$, $C_{Neg}$ and $C_{Neu}$. We identify these clusters in two different ways. In Table 6, we identify clusters for each trending topic in the query set. For a user's cluster membership identification,

**Table 5. Top 20 active users with their overall sentiment.**

| $I_1$ | $I_2$ | $I_3$ | $I_4$ | $I_5$ | $I_6$ | $I_7$ |
|---|---|---|---|---|---|---|
| js**an (0.262) | Ve**ar (-0.985) | PK**17 (-1.5) | ww**rr (-2.20) | wo**er (-1.843) | ze**ee (-0.533) | we**de (0.906) |
| ge**jo (-0.287) | ma**te (1.431) | tr**28 (0.222) | PK**17 (-2.7) | tr**28 (0.673) | wh**ra (-1.377) | Ve**ar (-0.16) |
| PK**17 (0.7989) | Pk**17 (0.441) | ra**99 (5.332) | mj**n7 (-2.050) | ki**59 (-0.886) | pu**v8 (-0.235) | HE**rk (0.37) |
| ev**67 (-4.746) | da**sa (0.221) | nu**me (0.650) | mi**k0 (0.475) | jg**00 (0.002) | me**at (0.183) | PK**17 (-5.4) |
| ca**oo (0.588) | br**om (0.642) | lg**87 (-2.051) | m2**30 (-3.171) | jb**28 (0.922) | ju**o2 (0.809) | da**y3 (-0.249) |
| bb**92 (-1.641) | Wa**ay (0.078) | jg**00 (0.534) | ki**59 (-0.844) | is**gs (-1.049) | ev**67 (0.602) | br**46 (-0.786) |
| ma**85 (-1.869) | RC**ic (1.116) | br**20 (-1.038) | js**vr (0.501) | go**bo (3.167) | ck**rz (-1.522) | Vi**rW (-0.398) |
| le**84 (1.239) | Ma**85 (-0.642) | Th**te (0.844) | jg**00 (1.068) | He**rk (0.011) | Ma**85 (-1.156) | US**aK (-1.194) |
| st**st (-0.6) | Jb**hn (-1.136) | Nu**60 (-2.599) | ex**99 (-0.407) | PK**17 (-4.2) | Ma**64 (-0.221) | TB**ne (-0.661) |
| gr**r6 (-2.728) | HE**rk (-0.442) | ma**te (-0.398) | br**20 (-2.449) | Ka**49 (0.108) | PK**17 (-5.1) | Sh**3R (-1.381) |
| ei**an (-0.618) | De**y_ (-0.393) | Ki**59 (-0.118) | SC**22 (1.639) | Jb**hn (-1.203) | Je**no (-0.429) | Nu**60 (1.057) |
| co**ly (0.921) | Ar**82 (-2.563) | Dj**a1 (0.840) | Ro**16 (-2.122) | Fi**ee (-0.553) | Fi**ee (-0.958) | Ma**85 (-2.478) |
| ch**ger (0.348) | 19**in (-0.685) | De**y_ (-0.329) | ka**49 (-0.753) | Cr**ef (0.365) | Eb**Jr (1.351) | Je**rs (-1.526) |
| be**ly (0.029) | wr**ub (-1.016) | Co**dy (-0.522) | Ha**99 (-0.797) | Ci**t1 (-0.739) | Dj**a1 (0.098) | Fl**x1 (-0.745) |
| cc**ly (0.458) | th**ia (-0.090) | 1f**at (0.859) | Cr**ef (0.365) | Ch**ger (-0.093) | De**ns (0.161) | De**ns (-1.20) |
| wr**ub (0.988) | js**vr (-0.759) | wr**ub (-1.016) | Ad**lG (-0.368) | Ap**ne (-3.092) | Ci**t1 (-1.840) | Bi**06 (-1.24) |
| ma**te (-1.839) | ev**67 (-3.682) | st**st (0.349) | Aa**ee (-0.263) | 24**ia (-0.661) | st**st (-2.2) | BF**in (1.548) |
| lo**79 (-1.471) | br**at (-1.344) | li**ze (-1.064) | zi**is (-1.043) | ra**an (0.043) | wo**er (-1.843) | 19**in (-2.913) |
| is**s (2.898) | br**20 (0.926) | ev**on (-1.303) | tr**28 (0.923) | ma**ra (-1.524) | ge**jo (1.2) | zi**is (-2.281) |
| ex**99 (-0.153) | Un**MN (2.203) | da**sh (-0.562 | ro**73 (-0.458) | lo**79 (1.194) | wa**27 (0.971) | ge**jo (-0.042) |

we sum up the sentiment scores of all tweets posted by that user on a particular topic. If she/he achieves a positive sentiment score, then he/she is the member of cluster $C_{Pos}$. For negative and neutral sentiment scores, a user is the member of $C_{Neg}$ and $C_{Neu}$ respectively. In all time intervals, the $C_{Neu}$ clusters have the highest number of members.

Next, we sum up each users' sentiment scores for top-$k = 3$ trending topics and consider these scores for clustering them. We determine $C_{Pos}$, $C_{Neg}$, and $C_{Neu}$ clusters following the same procedure of identifying topic-wise clusters. Table 7 represents the sizes of overall clusters at different time windows. Here, from the first time interval to the fifth time-interval, the positive, negative, and neutral clusters' size change typically. But, in the sixth and seventh time-interval, neutral cluster size increases than usual. From the analysis of this change, top-

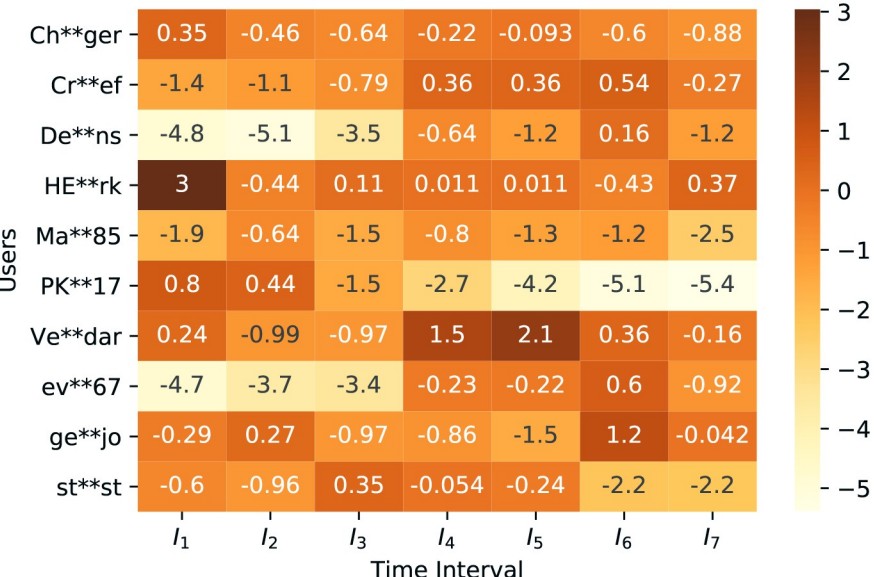

**Fig 7. Heatmap of selected ten users sentiment dynamics at each time-interval $I_m$ (shows the sentiments of selected ten users at different seven time-interval).**

$k = 3$ trending topics change that arises with Health, News and Polities topics. The UK's prime minister and health secretary test positive on 28 March 2020 that refers to the sixth time-interval. We sketch Fig 8 and visualize the changes in the clusters' size precisely for a graphical representation. The size of these clusters is changing with the shift of time windows. We also notice that the neutral clusters ($C_{Neu}$) always have the largest sizes among all.

As a first evaluation measure, we find out the entropy of our mentioned positive, negative, and neutral clusters. These clusters are shown in Table 8. Hence, a good sentimental cluster

**Table 6. Topic-wise sentimental clusters size at different $I_m$.**

| Time interval → | | $I_1$ | $I_2$ | $I_3$ | $I_4$ | $I_5$ | $I_6$ | $I_7$ |
|---|---|---|---|---|---|---|---|---|
| Clusters | Trending topics | | | | | | | |
| $C_{Pos}$ | Topic 1 | 592 (News) | 325 (News) | 324 (News) | 341 (News) | 332 (News) | 592 (Health) | 606 (Health) |
| | Topic 2 | 351 (Health) | 301 (Health) | 393 (Health) | 317 (Health) | 480 (Health) | 351 (News) | 357 (News) |
| | Topic 3 | 197 (Lock Down) | 253 (Covid Test) | 309 (Covid Test) | 281 (Covid Test) | 307 (Covid Test) | 197 (Politics) | 219 (Politics) |
| $C_{Neg}$ | Topic 1 | 644 (News) | 631 (News) | 642 (News) | 595 (News) | 673 (News) | 644 (Health) | 710 (Health) |
| | Topic 2 | 337 (Health) | 360 (Health) | 617 (Health) | 335 (Health) | 634 (Health) | 337 (News) | 367 (News) |
| | Topic 3 | 192 (Lock Down) | 308 (Covid Test) | 412 (Covid Test) | 293 (Covid Test) | 418 (Covid Test) | 192 (Politics) | 199 (Politics) |
| $C_{Neu}$ | Topic 1 | 973 (News) | 812 (News) | 995 (News) | 1001 (News) | 778 (News) | 912 (Health) | 927 (Health) |
| | Topic 2 | 931 (Health) | 775 (Health) | 948 (Health) | 977 (Health) | 981 (Health) | 903 (News) | 897 (News) |
| | Topic 3 | 907 (Lock Down) | 707 (Covid Test) | 893 (Covid Test) | 900 (Covid Test) | 705 (Covid Test) | 816 (Politics) | 878 (Politics) |

**Table 7. Overall sentimental clusters size for top-*k* = 3 trending topics.**

| Time Interval → | $I_1$ | $I_2$ | $I_3$ | $I_4$ | $I_5$ | $I_6$ | $I_7$ |
|---|---|---|---|---|---|---|---|
| Clusters ↓ | | | | | | | |
| $C_{Pos}$ | 987 | 809 | 1007 | 813 | 918 | 1073 | 1103 |
| $C_{Neg}$ | 1023 | 1109 | 1592 | 1211 | 1610 | 1192 | 1209 |
| $C_{Neu}$ | 1975 | 1821 | 1873 | 1993 | 1509 | 1912 | 1708 |

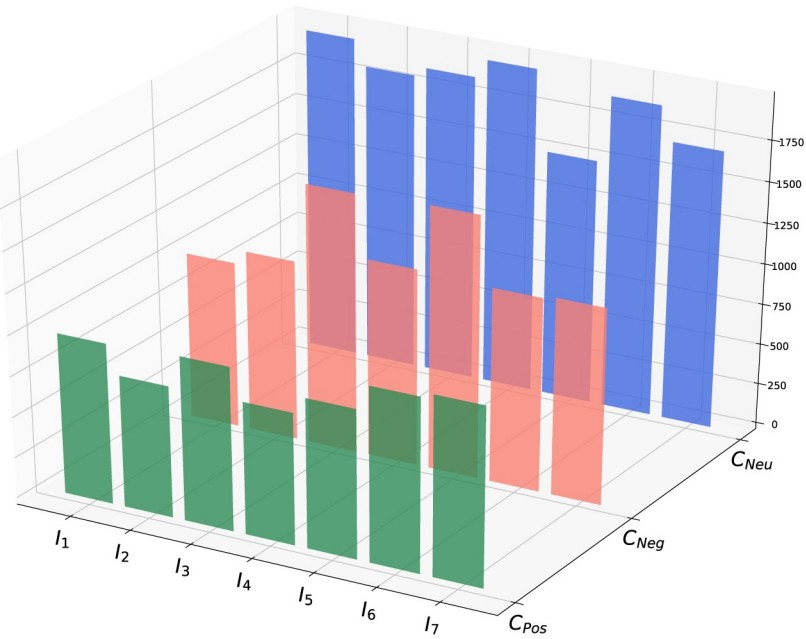

**Fig 8. Overall sentimental clusters at different time intervals $I_m$.**

should have a low entropy value, and here $C_{Neg}$ is 1.401 that depicts the lowest entropy value in the first time interval. The highest value of entropy is 1.584 as $C_{Pos}$ in the second time interval that refers to a bad sentimental cluster comparatively. We also see the diversity of entropy values where some values explicate good sentimental clusters, and some define bad sentimental clusters.

**Table 8. Entropy of sentimental clusters at different $I_m$.**

| Clusters → | $C_{Pos}$ | $C_{Neg}$ | $C_{Neu}$ |
|---|---|---|---|
| Time Interval ↓ | | | |
| $I_1$ | 1.437 | 1.401 | 1.53 |
| $I_2$ | 1.584 | 1.503 | 1.479 |
| $I_3$ | 1.579 | 1.563 | 1.58 |
| $I_4$ | 1.585 | 1.512 | 1.521 |
| $I_5$ | 1.548 | 1.561 | 1.497 |
| $I_6$ | 1.45 | 1.419 | 1.545 |
| $I_7$ | 1.465 | 1.402 | 1.46 |

**Table 9. Schematic cohesion of sentimental clusters at different $I_m$.**

| Time window | $C_{pos}$ | $C_{neg}$ | $C_{neu}$ |
|---|---|---|---|
| $I_1$ (23/3–25/3) | 0.60 (News) | 0.63 (News) | 0.493 (News) |
| $I_2$ (24/3–26/3) | 0.402 (News) | 0.569 (News) | 0.534 (News) |
| $I_3$ (25/3–27/3) | 0.39 (Health) | 0.403 (News) | 0.434 (News) |
| $I_4$ (26/3–28/3) | 0.419 (News) | 0.491 (News) | 0.502 (News) |
| $I_5$ (27/3–29/3) | 0.523 (Health) | 0.418 (News) | 0.518 (Health) |
| $I_6$ (28/3–30/3) | 0.552 (Health) | 0.54 (Health) | 0.472 (Health) |
| $I_7$ (29/3–31/3) | 0.549 (Health) | 0.587 (Health) | 0.525 (Health) |

Schematic cohesion, which is our second evaluation measure, is represented in Table 9 that leads to clusters' expertness. Here, we see the most outstanding value of schematic cohesion of $C_{pos}$ and $C_{neg}$ are 0.60 (News) and 0.63 (News) respectively in the first time interval. Furthermore, the most economical value of $C_{pos}$ is 0.39 (Health) in the third time interval. From the observation of this table, we see the heterogeneity among the clusters as values. Here, considering two topics, one is News, and another one is Health.

## 5 Discussion

Tracking top involved users' sentiments and sentimental clusters over time is the main objective of this work. Therefore, we conduct these experiments focusing on the topics that have the most trendiness on Twitter at a particular time.

Depending on the unique users' number and their activities on Twitter about a specific sub-topic related to COVID-19, we identify top-$k$ trending sub-topics. Table 1 holds information regarding trending sub-topics. Table 2 shows how the value of $\alpha$ controls given two parameters for a sub-topics trendiness detection. When we change the value of $\alpha$, the list of top trending subtopics is changing. It changes by either the topic title or by the serial of topics in the list. Notably, very few users can sustain the top involved list at all time intervals for related trendy topics. In Table 3 topic 'Lockdown' appears in the top sub-topics list at $I_1$ and then vanishes from the list after that. Other topics may remain on the list at more than a time window, but the percentages of their popularity change over time. This observation becomes clearer when we notice the heatmap in Fig 3. To find out the sentiment from the social stream, we use VADER. It depicts in Fig 9 as an architecture view. Table 4 shows some examples of social streams with sentiment results.

By using 'Users' involvement Detection Algorithm, we bring out top r involved users' sentiments and analysis over time. Table 5 holds the top 20 involved user's sentiment scores. Notably, very few users can sustain the top involved list at all time intervals for related trendy topics.

COVID-19 has a particular impact on users' sentiments. So we intend to focus on the most involved users' sentiments. With the flow of time, users' overall sentiments on top COVID-19 topics are changing. In Table 5, we can observe that the change of time window brings changes in the top 20 involved users' lists and their sentiments. This list in each $I_m$ is mixed with negative, positive, and neutral sentiments. Fig 7 has ten specific users' overall sentiment scores on

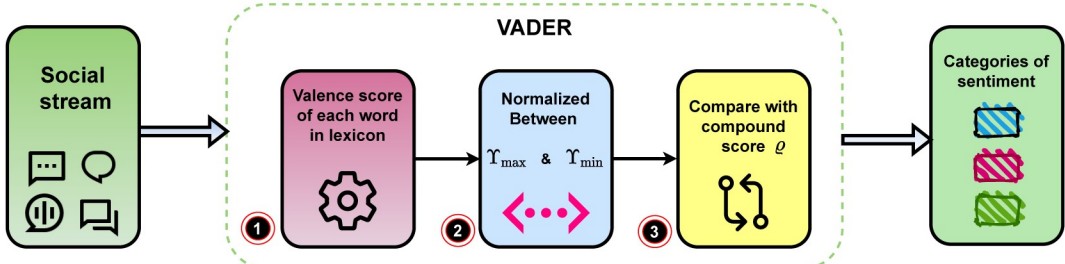

**Fig 9. Architecture of VADER.**

various time windows represented by a heat map that indicates these changes more specifically and visually.

We also illustrate the sentimental clusters topic-wise and overall. Table 6 shows the topic wise sentimental clusters and Table 7 displays the overall sentimental clusters in each time window. The 3D visualization can help to regulate the behavior of overall sentimental clusters. It is sketched in Fig 8.

Finally, Table 8 exhibits the entropy of clusters at each time window that serves the randomness of a cluster as the reference of entropy value. Table 9 depicts the schematic cohesion at each time window that mirrors the clusters' expertness.

## 6 Conclusion

Users' sentiment for diverse purposes has brought attention to research on social networks. It contains great importance in the COVID-19 pandemic situation. This paper proposed a model to identify users' sentiment dynamics for top-$k$ trending sub-topics related to COVID-19. It has also detected the top active users based on their involvement score on those trending topics.

This work successfully derives a function to calculates user's involvement scores towards Query topics and determines the top 20 involved users to analyze their sentiment at the different periods. We accomplish this research with the latest Twitter data and bring out that both users' involvement and their sentiments vary after a particular time. In the future, besides the determination of active users, we want to develop a methodology to track top negative and positive users by analyzing their sentiments.

## Author Contributions

**Data curation:** Md Shoaib Ahmed, Tanjim Taharat Aurpa.

**Formal analysis:** Tanjim Taharat Aurpa.

**Investigation:** Md Shoaib Ahmed.

**Methodology:** Md Shoaib Ahmed, Tanjim Taharat Aurpa.

**Supervision:** Md Musfique Anwar.

**Visualization:** Md Shoaib Ahmed.

**Writing – original draft:** Md Shoaib Ahmed, Tanjim Taharat Aurpa.

**Writing – review & editing:** Md Musfique Anwar.

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
