## [Decision Letter · Decision Letter 0]

2 Jun 2021

Detecting Sentiment Dynamics and Clusters of Twitter Users for Trending Topics in COVID-19 Pandemic

PONE-D-21-06706

Dear Dr. Ahmed,

We’re pleased to inform you that your manuscript has been judged scientifically suitable for publication and will be formally accepted for publication once it meets all outstanding technical requirements.

Kind regards,

Hocine Cherifi

Academic Editor

PLOS ONE

1. We suggest you thoroughly copyedit your manuscript for language usage, spelling, and grammar. If you do not know anyone who can help you do this, you may wish to consider employing a professional scientific editing service.  

Reviewers' comments:

Reviewer's Responses to Questions

**Comments to the Author**

1. Is the manuscript technically sound, and do the data support the conclusions?

Reviewer #1: Yes

2. Has the statistical analysis been performed appropriately and rigorously? 

Reviewer #1: Yes

3. Have the authors made all data underlying the findings in their manuscript fully available?

Reviewer #1: Yes

4. Is the manuscript presented in an intelligible fashion and written in standard English?

Reviewer #1: Yes

5. Review Comments to the Author

Reviewer #1: This is a very interesting research that studied the sentiment analysis with the emergent demand of covid related text. This research can help to identify the trending topics and find the related groups. It will help to fight the covid pandemic using data science techniques.

This work clearly presented the motivation, algorithms and validated the solution using the real datasets. The methodology and the experimental design are reasonable to show the result of such research.

There is only one minor comment: It missed the relevant citation - Detecting topic and sentiment dynamics due to Covid-19 pandemic using social media. H Yin, S Yang, J Li. International Conference on Advanced Data Mining and Applications, 610-623.

6. PLOS authors have the option to publish the peer review history of their article (what does this mean?). If published, this will include your full peer review and any attached files.

Reviewer #1: No

---

## [Editor Report · Acceptance letter]

30 Jul 2021

PONE-D-21-06706 

Detecting Sentiment Dynamics and Clusters of Twitter Users for Trending Topics in COVID-19 Pandemic 

Dear Dr. Ahmed:

I'm pleased to inform you that your manuscript has been deemed suitable for publication in PLOS ONE. Congratulations! Your manuscript is now with our production department. 

Kind regards, 

on behalf of

Professor Hocine Cherifi 

Academic Editor

PLOS ONE